# The Spectrum of the Fisher Information Matrix of a Single-Hidden-Layer Neural Network

**Jeffrey Pennington**
Google Brain
jpennin@google.com

**Pratik Worah**
Google Research
pworah@google.com

## Abstract

An important factor contributing to the success of deep learning has been the remarkable ability to optimize large neural networks using simple first-order optimization algorithms like stochastic gradient descent. While the efficiency of such methods depends crucially on the local curvature of the loss surface, very little is actually known about how this geometry depends on network architecture and hyperparameters. In this work, we extend a recently-developed framework for studying spectra of nonlinear random matrices to characterize an important measure of curvature, namely the eigenvalues of the Fisher information matrix. We focus on a single-hidden-layer neural network with Gaussian data and weights and provide an exact expression for the spectrum in the limit of infinite width. We find that linear networks suffer worse conditioning than nonlinear networks and that nonlinear networks are generically non-degenerate. We also predict and demonstrate empirically that by adjusting the nonlinearity, the spectrum can be tuned so as to improve the efficiency of first-order optimization methods.

## 1   Introduction

In recent years, the success of deep learning has spread from classical problems in image recognition [1], audio synthesis [2], translation [3], and speech recognition [4] to more diverse applications in unexpected areas such as protein structure prediction [5], quantum chemistry [5] and drug discovery [6]. These empirical successes continue to outpace the development of a concrete theoretical understanding of *how* and *in what contexts* deep learning works. A central difficulty in analyzing deep learning systems stems from the complexity of neural network loss surfaces, which are highly non-convex functions, often of millions or even billions [7] of parameters.

Optimization in such high-dimensional spaces poses many challenges. For most problems in deep learning, second-order methods are too costly to perform exactly. Despite recent developments on efficient approximations of these methods, such as the Neumann optimizer [8] and K-FAC [9], most practitioners use gradient descent and its variants [10], [11]. Despite their widespread use, it is not obvious why first-order methods are often successful in deep learning since it is known that first-order methods perform poorly in the presence of pathological curvature. An important open question in this direction is to what extent pathological curvature pervades deep learning and how it can be mitigated. More broadly, in order to continue improving neural network models and performance, we aim to better understand the conditions under which first-order methods will work well, and how those conditions depend on model design choices and hyperparameters.

Among the variety of objects that may be used to quantify the geometry of the loss surface, two matrices have elevated importance: the Hessian matrix and the Fisher information matrix. From the perspective of Euclidean coordinate space, the Hessian matrix is the natural object with which to quantify the local geometry of the loss surface. It is also the fundamental object underlying many second-order optimization schemes and its spectrum provides insights as to the nature of critical

points. From the perspective of information geometry, distances are measured in model space rather than in coordinate space, and the Fisher information matrix defines the metric and determines the update directions in natural gradient descent [12]. In contrast to the standard gradient, the natural gradient defines the direction in the parameter space which gives the largest change in the objective per unit change in the model, as measured by Kullback-Leibler divergence. As we discuss in Section 2, the Hessian and the Fisher are related; for the squared error loss functions that we consider in this work, it turns out that the Fisher equals the Gauss-Newton approximation of the Hessian, so the connection is concrete.

A central difficulty in building up a robust understanding of the properties of these curvature matrices stems from the fact that they are high-dimensional and the empirical estimation of their spectra is limited by memory and computational constraints. These limitations typically prevent direct calculations for models with more than a few tens of thousands of parameters and it is difficult to know whether conclusions drawn from such small models would generalize to the mega- or giga-dimensional networks used in practice.

It is therefore important to develop theoretical tools to analyze the spectra of these matrices. In general, the spectra will depend in intimate ways on the specific parameter values of the weights and the distribution of input data to the network. It is not feasible to precisely capture all of these details, and even if a theory were developed that did so, it would not be clear how to derive generalizable conclusions from it. We therefore focus on a simplified configuration in which the weights and inputs are taken to be random variables. The analysis then becomes a well-defined computation in random matrix theory.

The Fisher is a nonlinear function of the weights and data. To compute its spectrum, we extend the framework developed by Pennington and Worah [13] to study random matrices with nonlinear dependencies. As we describe in Section 2.4, the Fisher also has an internal block structure that complicates the resulting combinatorial analysis. The main technical contribution of this work is to extend the nonlinear random matrix theory of [13] to matrices with nontrivial internal structure.

The result of our analysis is an explicit characterization of the spectrum of the Fisher information matrix of a single-hidden-layer neural network with squared loss, random Gaussian weights and random Gaussian input data in the limit of large width. We draw several nontrivial and potentially surprising conclusions about the spectrum. For example, linear networks suffer worse conditioning than any nonlinear network, and although nonlinear networks may have many small eigenvalues they are generically non-degenerate. Our results also suggest precise ways to tune the nonlinearity in order to improve conditioning of the spectrum, and our empirical simulations show improvements in the speed of first-order optimization as a result.

## 2 Preliminaries

### 2.1 Notation and problem statement

Consider a single-hidden-layer neural network with weight matrices $W^{(1)}, W^{(2)} \in \mathbb{R}^{n \times n}$ and pointwise activation function $f : \mathbb{R} \to \mathbb{R}$. For input $X \in \mathbb{R}^n$, the output of the network $\hat{Y}(X) \in \mathbb{R}^n$ is given by $\hat{Y}(X) = W^{(2)} f(W^{(1)} X)$. For concreteness, we focus our analysis on the case of squared loss, in which case,

$$\mathcal{L}(\theta) = \mathbb{E}_{X,Y} \frac{1}{2} \|Y - \hat{Y}(X)\|_2^2, \tag{1}$$

where $Y \in \mathbb{R}^n$ are the regression targets and $\theta$ denotes the vector of all parameters $\{W^{(1)}, W^{(2)}\}$. The matrix of second derivatives or *Hessian* of the loss with respect to the parameters can be written as,

$$H = H^{(0)} + H^{(1)}, \tag{2}$$

where,

$$H_{ij}^{(0)} = \mathbb{E}_X \sum_\alpha \frac{\partial \hat{Y}_\alpha}{\partial \theta_i} \frac{\partial \hat{Y}_\alpha}{\partial \theta_j}, \quad \text{and} \quad H_{ij}^{(1)} = \mathbb{E}_X \sum_\alpha (\hat{Y}(X) - Y)_\alpha \frac{\partial^2 \hat{Y}_\alpha}{\partial \theta_i \partial \theta_j}. \tag{3}$$

In this work we focus on the positive-semi-definite matrix $H^{(0)}$, which is known as the Gauss-Newton matrix. It can also be written as $H^{(0)} = J^T J$, where $J \in \mathbb{R}^{n \times 2n^2}$ is the Jacobian matrix of $\hat{Y}$ with

respect to the parameters $\theta$. For models with squared loss, it is known that the Gauss-Newton matrix is equal to the Fisher information matrix of the model distribution with respect to its parameters [14]. As such, by studying $H^{(0)}$ we simultaneously examine the Gauss-Newton matrix and the Fisher information matrix.

The distribution of eigenvalues or *spectrum* of curvature matrices like $H^{(0)}$ plays an important role in optimization, as it characterizes the local geometry of the loss surface and the efficiency of first-order optimization methods. In this work, we seek to build a detailed understanding of this spectrum and how the architectural components of the neural network influence it. In order to isolate these factors from idiosyncratic behavior related to the specifics of the data and weight configurations, we focus on the a vanilla baseline configuration in which the data and the weights are both taken to be iid Gaussian random variables.

Concretely, we take $X \sim \mathcal{N}(0, I_n)$, $W_{ij}^{(l)} \sim \mathcal{N}(0, \frac{1}{n})$, and we will be interested in computing the expected distribution of eigenvalues $H^{(0)}$ for large $n$. From this perspective, the problem can be framed as a computation in random matrix theory, the principles behind which we now review.

## 2.2 Spectral density and the Stieltjes transform

The *empirical spectral density* of a matrix $M$ is defined as,

$$\rho_M(\lambda) = \frac{1}{n} \sum_{j=1}^{n} \delta(\lambda - \lambda_j(M)) , \tag{4}$$

where the $\lambda_j(M)$, $j = 1, \dots, n$, denote the $n$ eigenvalues of $M$, including multiplicity, and $\delta$ is the Dirac delta function. The *limiting spectral density* is the limit of eqn. (4) as $n \to \infty$, if it exists.

For $z \in \mathbb{C} \setminus \operatorname{supp}(\rho_M)$ the *Stieltjes transform* $G$ of $\rho_M$ is defined as,

$$G(z) = \int \frac{\rho_M(t)}{z - t} dt = -\frac{1}{n} \mathbb{E} \operatorname{tr}(M - zI_n)^{-1} , \tag{5}$$

where the expectation is with respect to the random variables $W$ and $X$. The quantity $(M - zI_{n_1})^{-1}$ is the *resolvent* of $M$. The spectral density can be recovered from the Stieltjes transform using the inversion formula,

$$\rho_M(\lambda) = -\frac{1}{\pi} \lim_{\epsilon \to 0^+} \operatorname{Im} G(\lambda + i\epsilon) . \tag{6}$$

## 2.3 Moment method

One of the main tools for computing the limiting spectral distributions of random matrices is the moment method, which, as the name suggests, is based on computations of the moments of $\rho_M$. The asymptotic expansion of eqn. (5) for large $z$ gives the Laurent series,

$$G(z) = \sum_{k=0}^{\infty} \frac{m_k}{z^{k+1}} , \tag{7}$$

where $m_k$ is the $k$th moment of the distribution $\rho_M$,

$$m_k = \int dt \, \rho_M(t) t^k = \frac{1}{n} \mathbb{E} \operatorname{tr} M^k . \tag{8}$$

If one can compute $m_k$, then the density $\rho_M$ can be obtained via eqns. (7) and (6). The idea behind the moment method is to compute $m_k$ by expanding out powers of $M$ inside the trace as,

$$\frac{1}{n} \mathbb{E} \operatorname{tr} M^k = \frac{1}{n} \mathbb{E} \sum_{i_1, \dots, i_k \in [n]} M_{i_1 i_2} M_{i_2 i_3} \cdots M_{i_{k-1} i_k} M_{i_k i_1} , \tag{9}$$

and evaluating the leading contributions to the sum as $n \to \infty$. We will use the moment method in order to compute the limiting spectral density of the Fisher. As a first step in that direction, we focus on the properties of the layer-wise block structure in the Fisher induced by the neural network architecture.

## 2.4 Block structure of the Fisher

As described above, in our single-hidden-layer setting with squared loss, the Fisher is given by

$$H^{(0)} = \mathbb{E}_X \left[ J^T J \right], \quad J_{\alpha i} = \frac{\partial \hat{Y}_\alpha}{\partial \theta_i}. \tag{10}$$

Because the parameters of the model are organized into two layers, it is convenient to partition the Fisher into a $2 \times 2$ block matrix,

$$H^{(0)} = \begin{pmatrix} H_{11}^{(0)} & H_{12}^{(0)} \\ H_{12}^{(0)T} & H_{22}^{(0)} \end{pmatrix}.$$

Furthermore, because the parameters of each layer are matrices, it is useful to regard each block of the Fisher as a four-index tensor. In particular,

$$[H_{11}^{(0)}]_{a_1 b_1, a_2 b_2} = \mathbb{E}_X \left[ \sum_i J_{i,a_1 b_1}^{(1)} J_{i,a_2 b_2}^{(1)} \right],$$

$$[H_{12}^{(0)}]_{a_1 b_1, c_1 d_1} = \mathbb{E}_X \left[ \sum_i J_{i,a_1 b_1}^{(1)} J_{i,c_1 d_1}^{(2)} \right],$$

$$[H_{22}^{(0)}]_{c_1 d_1, c_2 d_2} = \mathbb{E}_X \left[ \sum_i J_{i,c_1 d_1}^{(2)} J_{i,c_2 d_2}^{(2)} \right].$$

The Jacobian entries $J_{i,ab}^{(l)}$ equal the derivatives of $\hat{Y}_i$ with respect to the weight variables $W_{ab}^{(l)}$,

$$J_{i,ab}^{(1)} = W_{ia}^{(2)} f'\left(\sum_k W_{ak}^{(1)} X_k\right) X_b, \qquad J_{j,cd}^{(2)} = \delta_{cj} f\left(\sum_l W_{dl}^{(1)} X_l\right), \tag{11}$$

where $\delta_{cj}$ denotes the Kronecker delta function i.e., it is 1 if $c = j$, and 0 otherwise.

In order to proceed by the method of moments, we will need to compute the normalized trace of powers of the Fisher, i.e. $\frac{1}{n} \text{tr}[H^{(0)}]^d$, for any $d$. The block structure of the Fisher makes the explicit representation of these traces somewhat complicated. The following proposition helps simplify the expressions.

**Proposition 1.** *Let* $A_1 \in \mathbb{R}^{n \times k_1}$, $A_2 \in \mathbb{R}^{n \times k_2}$ *and* $A = [A_1, A_2] \in \mathbb{R}^{n \times (k_1 + k_2)}$. *Then,*

$$\text{tr}[(A^T A)^d] = \sum_{b \in \{1,2\}^d} \text{tr} \prod_{i=1}^d A_{b_i} A_{b_i}^T = \sum_{b \in \{1,2\}^d} \text{tr} \, A_{b_d}^T A_{b_1} \prod_{i=1}^{d-1} A_{b_i}^T A_{b_{i+1}}. \tag{12}$$

Using Proposition 1 with $A_1 = J^{(1)}$ and $A_2 = J^{(2)}$, we have,

$$\text{tr}[(H^{(0)})^d] = \sum_{b \in \{1,2\}^d} \text{tr} \, \mathbb{E}_X \left[ J^{(b_d)T} J^{(b_1)} \right] \prod_{i=1}^{d-1} \mathbb{E}_X \left[ J^{(b_i)T} J^{(b_{i+1})} \right], \tag{13}$$

which expresses the traces of the block Fisher entirely in terms of products of its constituent blocks.

In order to carry out the moment method to completion, we need the expected normalized traces $m_k$,

$$m_k = \frac{1}{n} \mathbb{E}_W \, \text{tr}[(H^{(0)})^k], \tag{14}$$

in the limit of large $n$. Because the nonlinearity significantly complicates the analysis, we first illustrate the basics of the methodology in the linear case before moving on to the general case.

## 2.5 An Illustrative Example: The Linear Case

Let us assume that $f$ is the identity function i.e., $f(z) = z$. In this case, eqn. (11) can be written as,

$$J^{(1)} = W^{(2)T} \otimes X, \qquad J^{(2)} = I \otimes W^{(1)} X. \tag{15}$$

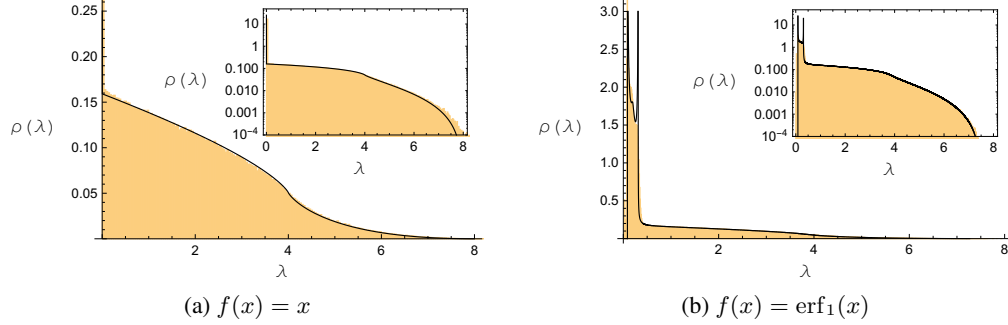

(a) $f(x) = x$        (b) $f(x) = \mathrm{erf}_1(x)$

Figure 1: Empirical spectra of Fisher for single-hidden-layer networks of width 128 (orange) and theoretical prediction of spectra (black) for (a) linear and (b) $\mathrm{erf}_1$ (see eqn. 30) networks. Insets show logarithmic scale.

Using the fact that $\mathbb{E}_X[XX^T] = I_n$, eqn. (13) gives,

$$\mathrm{tr}\,[(H^{(0)})^d] = \mathbb{E}_W \sum_{k=0}^{d} \binom{d}{k} \mathrm{tr}(W^{(2)}W^{(2)T})^{d-k} \mathrm{tr}(W^{(1)}W^{(1)T})^k = \sum_{k=0}^{d} \binom{d}{k} C_{d-k}C_k\,, \quad (16)$$

where $C_n$ is the $n$th Catalan number. The series can be summed to obtain the Stieltjes transform, whose imaginary part gives the following explicit form for the spectrum,

$$\rho(\lambda) = \frac{1}{2}\delta(\lambda) + \left[\frac{1}{2\pi^2}E\Big(\frac{1}{16}(8-\lambda)\lambda\Big) + \frac{4-\lambda}{8\pi^2}K\Big(\frac{1}{16}(8-\lambda)\lambda\Big)\right]\mathbf{1}_{[0,8]}\,, \quad (17)$$

where $K$ and $E$ are the complete elliptic integrals of the first- and second-kind,

$$K(k) = \int_0^{\frac{\pi}{2}} d\theta\, \frac{1}{\sqrt{1-k\sin^2\theta}}\,, \quad E(k) = \int_0^{\frac{\pi}{2}} d\theta\, \sqrt{1-k\sin^2\theta}\,. \quad (18)$$

Notice that the spectrum is highly degenerate, with half of the eigenvalues equaling zero. This degeneracy can be attributed to the $GL(n^2)$ symmetry of the product $W^{(2)}W^{(1)}$ under $\{W^{(1)}, W^{(2)}\} \to \{GW^{(1)}, W^{(2)}G^{-1}\}$. Fig. 1a shows excellent agreement between the predicted spectral density and finite-width empirical simulations.

## 3 The Stieltjes transform of $H^{(0)}$

### 3.1 Main Result

If $f : \mathbb{R} \to \mathbb{R}$ is an activation function with zero Gaussian mean and finite Gaussian moments,

$$\int \frac{dx}{\sqrt{2\pi}}\, e^{-\frac{x^2}{2}} f(x) = 0\,, \qquad \left|\int \frac{dx}{\sqrt{2\pi}}\, e^{-\frac{x^2}{2}} f(x)^k\right| < \infty\,, \text{ for } k > 1\,, \quad (19)$$

then the Stieltjes transform of the limiting spectral density of $H^{(0)}$ is given by the following theorem.

**Theorem 1.** *The Stieltjes transform of the spectral density of the Fisher information matrix of a single-hidden-layer neural network with squared loss, activation function $f$, weight matrices $W^{(1)}, W^{(2)} \in \mathbb{R}^{n \times n}$ with iid entries $W_{ij}^{(l)} \sim \mathcal{N}(0, \frac{1}{n})$, no biases, and iid inputs $X \sim \mathcal{N}(0, I_n)$ is given by the following integral as $n \to \infty$:*

$$G(z) = \int_{\mathbb{R}} \int_{\mathbb{R}} \frac{\lambda_1 + \lambda_2 - 2z}{2\zeta^2\big((\eta-\zeta)(\eta'-\zeta) + \lambda_1(z-\eta+\zeta) + \lambda_2(z-\eta'+\zeta) - z^2\big)} d\mu_1(\lambda_1)d\mu_2(\lambda_2)\,, \tag{20}$$

*where the constants $\eta$, $\eta'$, and $\zeta$ are determined by the nonlinearity,*

$$\eta = \int_{\mathbb{R}} f(x)^2 \frac{e^{-x^2/2}}{\sqrt{2\pi}}dx\,, \quad \eta' = \int_{\mathbb{R}} f'(x)^2 \frac{e^{-x^2/2}}{\sqrt{2\pi}}dx\,, \quad \zeta = \left(\int_{\mathbb{R}} f'(x)\frac{e^{-x^2/2}}{\sqrt{2\pi}}dx\right)^2\,, \quad (21)$$

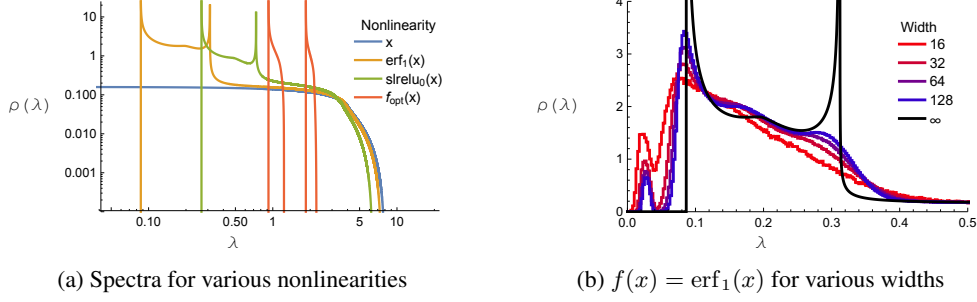

(a) Spectra for various nonlinearities

(b) $f(x) = \mathrm{erf}_1(x)$ for various widths

Figure 2: (a) Theoretical predictions for spectra of various nonlinearities; see eqns. (28) and (30). The linear case is degenerate and more poorly conditioned than the nonlinear cases. (b) Theoretical prediction of spectrum for $\mathrm{erf}_1$ compared with empirical simulations. Practical constraints restrict the width to small values, but slow convergence toward the asymptotic prediction can be observed.

*and the measures $d\mu_1$ and $d\mu_2$ are given by,*

$$d\mu_1(\lambda_1) = \frac{1}{2\pi}\sqrt{\frac{\eta' + 3\zeta - \lambda_1}{\lambda_1 - \eta' + \zeta}}\,\mathbf{1}_{[\eta'-\zeta,\,\eta'+3\zeta]}, \quad d\mu_2(\lambda_2) = \frac{1}{2\pi}\sqrt{\frac{\eta + 3\zeta - \lambda_2}{\lambda_2 - \eta + \zeta}}\,\mathbf{1}_{[\eta-\zeta,\,\eta+3\zeta]}. \quad (22)$$

**Remark 1.** *A straightforward application of Carlson's algorithm [15] can reduce the integral in eqn. (20) to a combination of three standard elliptic integrals.*

**Remark 2.** *The spectral density can be recovered from eqn. (20) through the inversion formula, eqn. (6).*

**Remark 3.** *Although the result in Theorem 1 is written in terms of $f'$, it is not necessary that $f$ be differentiable. In fact, the weak derivative can be used in place of the derivative, as the proof of the reduction (see also [13]) to final form uses integration by parts only. Therefore, just the existence of a weak derivative for $f$ suffices. In particular, the result would hold for $|x|$ and Relu functions.*

The proof of Theorem 1 is quite long and technical, so it's deferred to the Supplementary Material. The basic idea underlying the proof is very similar to that utilized in [13]. The calculation of the moments is divided into two sub-problems, one of enumerating certain connected outer-planar graphs, and another of evaluating certain high-dimensional integrals that correspond to walks in those graphs.

Fig. 1 shows the excellent agreement of the predicted spectrum with empirical simulations of finite-width networks. Fig. 2 highlights the region of the spectrum for which the asymptotic behavior is slow to set in and suggests that empirical simulations with small networks may not provide an accurate portrayal of the behavior of large networks. Fig. 2a shows the predicted spectra for a variety of nonlinearities.

### 3.2 Features of the spectrum

Owing to eqn. (6), the branch points and poles of $G(z)$ encode information about the delta function peaks, spectral edges, and discontinuities in the derivative of $\rho(\lambda)$. These special points can be determined directly from the integral representation for $G(z)$ in eqn. (20) by examining the zeros of the denominator of the integrand. In particular, the following six values of $z$ are locations of the poles at the integration endpoints and determine the salient features of the spectral density:

$$z_1 = \eta - \zeta, \quad z_2 = \eta + 3\zeta, \quad z_3 = \frac{1}{2}\left(\eta + \eta' + 6\zeta - \sqrt{(\eta' - \eta)^2 + 64\zeta^2}\right), \quad (23)$$

$$z_4 = \eta' - \zeta, \quad z_5 = \eta' + 3\zeta, \quad z_6 = \frac{1}{2}\left(\eta + \eta' + 6\zeta + \sqrt{(\eta' - \eta)^2 + 64\zeta^2}\right). \quad (24)$$

In the Supplementary Material, we establish the relative ordering of constants $0 \le \zeta \le \eta \le \eta'$, which implies that the minimum and maximum eigenvalues of $H^{(0)}$ are given by,

$$\lambda_{\min} = z_1, \quad \text{and} \quad \lambda_{\max} = z_6. \quad (25)$$

Table 1: Properties of nonlinearities

| | $\eta$ | $\eta'$ | $\zeta$ | | | Locations of spectral features | | | |
| | | | | $z_1$ | $z_2$ | $z_3$ | $z_4$ | $z_5$ | $z_6$ |
|---|---|---|---|---|---|---|---|---|---|
| $x$ | 1 | 1 | 1 | 0 | 4 | 0 | 0 | 4 | 8 |
| $\mathrm{erf}_1(x)$ | 1 | 1.226 | 0.914 | 0.086 | 3.741 | 0.198 | 0.312 | 3.966 | 7.51 |
| $\mathrm{srelu}_0(x)$ | 1 | 1.467 | 0.733 | 0.267 | 3.200 | 0.491 | 0.733 | 3.667 | 6.377 |
| $f_{\mathrm{opt}}$ | 1 | 1.923 | 0.077 | 0.923 | 1.231 | 1.138 | 1.846 | 2.154 | 2.247 |

The Supplementary Material also shows that the equality $\eta = \zeta$ only holds for linear networks, which implies that the minimum eigenvalue is nonzero for every nonlinear activation function. There are two delta function peaks in spectrum, which are located at,

$$\lambda_{\mathrm{peak}}^{(1)} = \lambda_{\min} = z_1, \quad \text{and} \quad \lambda_{\mathrm{peak}}^{(2)} = z_4. \tag{26}$$

These peaks indicate specific eigenvalues that have nonvanishing probability of occurrence. These peaks coalesce when $\eta = \eta'$, which can only happen for linear activation functions, in which case $\eta = \eta' = \zeta$, so the peaks occur at $\lambda = 0$, as illustrated in Fig. 2a. That figure also shows that the spectrum may consist of two disconnected components, in which case $z_2$ is the location of the right edge of the left component. Finally, the derivative of the spectrum is discontinuous at $z_3$ and $z_5$. These predictions can be verified in Fig. 2a by consulting Table 1, which provides numerical values for these special points for the various nonlinearities appearing in the figure.

## 4 Empirical analysis

### 4.1 A measure of conditioning

Using the results from Section 4.1, the first two moments can be given explicitly as,

$$
\begin{aligned}
m_1 &= \lim_{n \to \infty} \frac{1}{n} \mathrm{tr}[H^{(0)}] = \frac{1}{2}(\eta + \eta') \\
m_2 &= \lim_{n \to \infty} \frac{1}{n} \mathrm{tr}[H^{(0)2}] = \frac{1}{2}(\eta^2 + \eta'^2 + 4\zeta^2)
\end{aligned}
\tag{27}
$$

A scale-invariant measure of conditioning of the Fisher is just $m_2/m_1^2$, which is lower-bounded by 1, and which quantifies how tightly concentrated the spectrum is around its mean. Ideally, this quantity should be as small as possible to avoid pathological curvature and to enable fast first-order optimization. One advantage of $m_2/m_1^2$ compared to other condition numbers such as $\lambda_{\max}/\lambda_{\min}$ or $\lambda_{\max}$ is that it is scale-invariant and well-defined even in the presence of degeneracy in the spectrum.

By expanding $f$ in a basis of Hermite polynomials, we show in the Supplementary Material that among the functions with zero Gaussian mean that

$$f_{\mathrm{opt}}(x) = \frac{1}{\sqrt{13}}\left(x + \sqrt{6}(x^2 - 1)\right) \tag{28}$$

minimizes the ratio $m_2/m_1^2$. Note that we have removed the freedom to rescale $f_{\mathrm{opt}}$ by a constant by enforcing $\eta = 1$. Curiously, a linear activation function actually maximizes the ratio, implying that nonlinearity invariably improves conditioning, at least by this measure. The relative conditioning of spectra resulting from various activation functions can be observed in Fig. 2a.

The function $f_{\mathrm{opt}}(x)$ grows quickly for large $|x|$ and may be too unstable to use in actual neural networks. Alternative functions could be found by solving the optimization problem,

$$f_* = \arg\min_f \frac{m_2}{m_1^2}, \tag{29}$$

subject to some constraints, for example that $f$ be monotone increasing, have zero Gaussian mean, and saturate for large $|x|$. Such a problem could be solved via variational calculus; see the Supplementary Material.

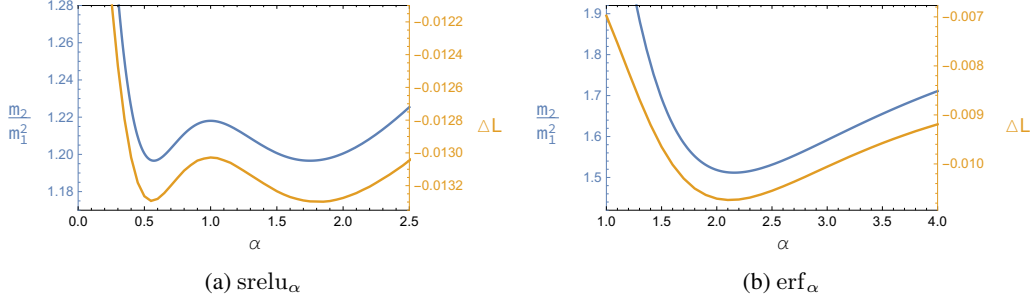

<div align="center">(a) $\mathrm{srelu}_\alpha$          (b) $\mathrm{erf}_\alpha$</div>

Figure 3: Comparison of the conditioning measure $m_2/m_1^2$ and single-step loss reduction $\Delta L$ (eqn. (33)) as the activation function changes for (a) $\mathrm{srelu}_\alpha$ and (b) $\mathrm{erf}_\alpha$ (eqn. (30)). The curves are highly correlated, suggesting the possibility of improved first-order optimization performance by tuning the spectrum of the Fisher through the choice of activation function.

## 4.2 Efficiency of gradient descent

Another way to investigate the ratio $m_2/m_1^2$ is to see how well it correlates with the efficiency of first-order optimization. For this purpose, we examine two one-parameter classes of well-behaved activation functions related to ReLU and the error function,

$$\mathrm{srelu}_\alpha(x) = \frac{[x]_+ + \alpha[-x]_+ - \frac{1+\alpha}{\sqrt{2\pi}}}{\sqrt{\frac{1}{2}(1+\alpha^2) - \frac{1}{2\pi}(1+\alpha)^2}}, \qquad \mathrm{erf}_\alpha(x) = \frac{\mathrm{erf}(\alpha^2 x)}{\sqrt{\frac{4}{\pi}\tan^{-1}\sqrt{1+4\alpha^4} - 1}}. \tag{30}$$

Here $\mathrm{srelu}_\alpha$ is the shifted leaky ReLU function studied in [13]. Both $\mathrm{srelu}_\alpha$ and $\mathrm{erf}_\alpha$ have zero Gaussian mean and are normalized such that $\eta = 1$ for all $\alpha$. Changing $\alpha$ does affect $\eta'$, $\zeta$ and the ratio $m_2/m_1^2$, which implies that different functions within these one-parameter families may behave quite differently under gradient descent.

We designed a simple and controlled experiment to explore these differences in the context of neural network training. The setup is a modified student-teacher framework, in which the student is initialized with the teacher's parameters, but the regression targets are perturbed so that student's parameters are suboptimal. Then we ask by how much can the student decrease the loss by one optimally-chosen step in the gradient direction. Concretely, we define

$$Y_i = W_t^{(2)} f(W_t^{(1)} X_i) + \epsilon_i, \qquad i = 1, \dots, M, \tag{31}$$

for teacher weights $[W_t^{(l)}]_{ij} \sim \mathcal{N}(0, \frac{1}{n})$, $X_i \sim \mathcal{N}(0, I_n)$, and $\epsilon_i \sim \mathcal{N}(0, \varepsilon^2 I_n)$, with width $n = 2^7$, number of samples $M = 2^{17}$, and perturbation size $\varepsilon = 10^{-3}$. The loss is defined as,

$$L(W_s) = \sum_{i=1}^{M} \frac{1}{2} \|Y_i - W_s^{(2)} f(W_s^{(1)} X_i)\|_2^2. \tag{32}$$

We are interested in the maximal single-step loss decrease when $W_s$ is initialized at $W_t$, i.e.,

$$\Delta L = \min_\eta \left[ L(W_t - \eta \nabla L|_{W_t}) - L(W_t) \right]. \tag{33}$$

For the two classes of activation functions in eqn. (30), we empirically measured $\Delta L$ as a function of $\alpha$. In Fig. 3 we compare the results with our theoretical predictions for $m_2/m_1^2$ as a function of $\alpha$. The agreement is excellent, suggesting that our theory may be able to make practical predictions regarding training efficiency of actual neural networks.

## 5 Conclusions

In this work, we computed the spectrum of the Fisher information matrix of a single-hidden-layer neural network with squared loss and Gaussian weights and Gaussian data in the limit of large network width. Our explicit results indicate that linear networks suffer worse conditioning than

nonlinear networks and that although nonlinear networks may have numerous small eigenvalues they are generically non-degenerate. We also showed that by tuning the nonlinearity it is possible to adjust the spectrum in such a way that the efficiency of first-order optimization methods can be improved. By undertaking this analysis, we demonstrated how to extend the techniques developed in [13] for studying random matrices with nonlinear dependencies to the block-structured curvature matrices that are relevant for optimization in deep learning. The techniques presented here pave the way for future work studying deep learning via random matrix theory.

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
