[Supplementary Material]

# Supplemental Material:
## The Spectrum of the Fisher Information Matrix of a Single-Hidden-Layer Neural Network

## 1 Hermite expansion

Any function with finite Gaussian moments can be expanded in a basis of Hermite polynomials. Defining

$$H_n(x) = (-1)^n e^{\frac{x^2}{2}} \frac{\partial^n}{\partial x^n} e^{-\frac{x^2}{2}} \tag{S1}$$

we can write,

$$f(x) = \sum_{n=0}^{\infty} \frac{f_n}{\sqrt{n!}} H_n(x), \tag{S2}$$

for some constants $f_n$. Owing the orthogonality of the Hermite polynomials, this representation is useful for evaluating Gaussian integrals. In particular, the condition that $f$ be centered is equivalent the vanishing of $f_0$,

$$0 = \int_{-\infty}^{\infty} dx \frac{e^{-x^2/2}}{\sqrt{2\pi}} f(x) \tag{S3}$$
$$= f_0.$$

The constants $\eta$, $\eta'$, and $\zeta$ are also easily expressed in terms of the coefficients,

$$\zeta = \left[ \int_{-\infty}^{\infty} dx \frac{e^{-x^2/2}}{\sqrt{2\pi}} f'(x) \right]^2 = f_1^2$$

$$\eta = \int_{-\infty}^{\infty} dx \frac{e^{-x^2/2}}{\sqrt{2\pi}} f(x)^2 = \sum_{n=0}^{\infty} f_n^2 = \zeta + \sum_{n=2}^{\infty} f_n^2 \equiv \zeta + r_1^2 \tag{S4}$$

$$\eta' = \int_{-\infty}^{\infty} dx \frac{e^{-x^2/2}}{\sqrt{2\pi}} f'(x)^2 = \sum_{n=0}^{\infty} n f_n^2 \equiv \zeta + 2r_1^2 + r_2^2,$$

where,

$$r_1^2 = \sum_{n=2}^{\infty} f_n^2, \quad r_2^2 = \sum_{n=3}^{\infty} (n-2) f_n^2. \tag{S5}$$

From this representation it is easy to see that $0 \le \zeta \le \eta \le \eta'$. The first and second moments of the Fisher are then given by,

$$m_1 = \frac{1}{2}(\eta + \eta') = \frac{1}{2}(2\zeta + 3r_1^2 + r_2^2)$$
$$m_2 = \frac{1}{2}(\eta^2 + \eta'^2 + 4\zeta^2) = \frac{1}{2}(6\zeta^2 + 6r_1^2\zeta + 2r_2^2\zeta + 5r_1^4 + 4r_1^2 r_2^2 + r_2^4). \tag{S6}$$

Viewed as a function of $\zeta$, $r_1$, and $r_2$, the ratio $m_2/m_1^2$ has three critical points over the positive real numbers,

$$\text{(a) } r_1 = 0, \quad r_2 = 0, \quad \frac{m_2}{m_1^2} = 3 \tag{S7}$$

$$\text{(b) } r_1 = 0, \quad r_2^2 = 4\zeta, \quad \frac{m_2}{m_1^2} = \frac{5}{3} \tag{S8}$$

$$\text{(c) } r_1^2 = 12\zeta, \quad r_2 = 0, \quad \frac{m_2}{m_1^2} = \frac{21}{19}. \tag{S9}$$

Solution (c) is the minimum. Note that $r_2 = 0$ implies $f_k = 0$ for $k \ge 3$. Without loss of generality we can set $\eta = 1$, in which case,

$$f_{\text{opt}}(x) = \frac{1}{\sqrt{13}}(x + \sqrt{6}(x^2 - 1)). \tag{S10}$$

## 2 Variational Calculus

It is not easy to systematically generalize the calculations in the previous section for arbitrary constraints on $f$. For example, placing lower bounds on $f$, requiring that $f$ is monotone, placing bounds on the derivative of $f$, and any other such conditions that are typically seen for activation functions are all valid additional constraints to compute a good conditioned $f$. Variational calculus is more suited for such a generalization. However, unlike the above explicit solution, the output is a PDE with boundary conditions. For example, adding constraints on the first derivatives beyond the above, only adds non-holonomic constraints to the problem [16]. Below, we outline the same calculation leading to the PDE necessarily satisfied by any extremal $f$.

Without loss of generality, let $\eta + \eta' = 1$, then the optimization problem becomes

$$f_* = \arg\min_f \frac{1}{2}\left(1 - 2\eta\eta' + 4\zeta^2\right), \qquad \text{s.t.} \quad \eta + \eta' = 1. \tag{S11}$$

Writing the $\zeta^2$ term as $\int F(x_1, x_2, x_3, x_4)d\vec{x}$, where

$$F(x_1, x_2, x_3, x_4) := \kappa \prod_{i=1}^{4} f(x_i) \frac{e^{-x_i^2/2}}{\sqrt{2\pi}}. \tag{S12}$$

Here $\kappa$ is a function of $x_i$s that is sharply concentrated around the line $x_1 = x_2 = x_3 = x_4$. The sharper the concentration of $\kappa$ around the line, the closer the solution is to the optimum value.

We have effectively reduced the polynomial objective function in $x$ to a multi-dimensional linear integral function in $x_i$s:

$$\min_f \left(1 - 2\int_{\mathbb{R}^2} f(x_1)^2 f'(x_2)^2 \frac{e^{-(x_1^2+x_2^2)/2}}{2\pi}d\vec{x} + 4\int_{\mathbb{R}^4} F(x_1, x_2, x_3, x_4)d\vec{x}\right), \tag{S13}$$

where each $x_i$ follows a isoperimetric constraint of the form:

$$\int_{\mathbb{R}} (f(x)^2 + f'(x)^2) \frac{e^{-x^2/2}}{\sqrt{2\pi}}dx = 1. \tag{S14}$$

Moreover, we have the (isoperimetric) "centering" constraint:

$$\int_{\mathbb{R}} f(x) \frac{e^{-x^2/2}}{\sqrt{2\pi}}dx = 0. \tag{S15}$$

Eqns. (S13), (S14) and (S15) lead to a standard Euler-Lagrange PDE, which can be simplified from symmetry considerations into an ODE, in the case above. See for example, Chapter 2 in [16]. The solution of that PDE gives the necessary condition, which is also usually sufficient, for the optimal activation function $f$.

## 3 Moments

Our main technical result will be stated in terms of generating functions arising from certain combinatorial settings, which are variants of certain standard problems and interesting by themselves. We introduce the following notation. Given a generating function $c(t) = \sum_k c_k t^k$, we denote $[c(t)]_i := c_i$ and for $c(t_1, t_2) = \sum_{k_1, k_2} c_{k_1, k_2} t_1^{k_1} t_2^{k_2}$, we denote $[c(t_1, t_2)]_{i,j} := c_{i,j}$.

**Lemma 1.** *The Stieltjes transform $G(z)$ of the spectral density of the Fisher information matrix of a single-hidden-layer neural network with squared loss, activation function $f$, weight matrices $W^{(1)}, W^{(2)} \in \mathbb{R}^{n \times n}$ with i.i.d. entries $W_{ij}^{(l)} \sim \mathcal{N}(0, \frac{1}{n})$, no biases, and i.i.d. inputs $X \sim \mathcal{N}(0, I_n)$ is given by the following integral as $n \to \infty$:*

$$G(z) = \frac{1}{z}P(\frac{1}{z}), \tag{S16}$$

*where the function $P(t)$ is given by the following series:*

$$P(t) = \frac{P(t; \eta, \zeta)}{2} + \frac{P(t; , \eta', \zeta)}{2} + \frac{1}{2} \sum_{n=1}^{\infty} \sum_{\substack{n_1, n_2 \\ n_1 + n_2 = n}} \sum_{d=1}^{\infty} [H(d, \lambda_1, \lambda_2)]_{n_1, n_2} [P_1(d, t)]_{n_1} [P_2(d, t)]_{n_2} t^{n+2(d-1)},$$

(S17)

*The generating functions $H$, with formal variables $\lambda_1$ and $\lambda_2$, is defined as follows:*

$$H(d, \lambda_1, \lambda_2) = \frac{2\lambda_1\lambda_2 - \lambda_1^2\lambda_2 - \lambda_1\lambda_2^2}{(-1+\lambda_1)^{d+1}(-1+\lambda_2)^{d+1}}.$$

(S18)

*The generating functions $P_1(d,t)$ and $P_2(d,t)$ can be characterized in terms of the generating function $P(t; \cdot, \cdot)$ obtained in the paper [13]:*

$$P_1(d, t) := \quad \zeta^{d-1} \sum_{i=0}^{d-1} \frac{\left( \frac{d-i-1}{d+i-1} \binom{d+i-1}{i} \right)}{(1 - P(t; \eta', \zeta))^{d-i-1}}$$

(S19)

$$P_2(d, t) := \quad \zeta^{d-1} \sum_{i=0}^{d-1} \frac{\left( \frac{d-i-1}{d+i-1} \binom{d+i-1}{i} \right)}{(1 - P(t; \eta, \zeta))^{d-i-1}},$$

(S20)

*where the generating function $P(t; \theta_1, \theta_2)$ with parameters $\theta_1$ and $\theta_2$ is given by the quadratic recurrence:*

$$P(t; \theta_1, \theta_2) = 1 + P(t; \theta_1, \theta_2)(\theta_1 - \theta_2)t + \frac{P(t; \theta_1, \theta_2)\theta_2 t}{1 - P(t; \theta_1, \theta_2)\theta_2 t}.$$

(S21)

**Remark 4.** *The significance of $P(t)$ in [13] is that it completely characterizes the (Stieltjes transform of) the singular values of the resolvent of the matrix $f(WX)$ i.e., the output obtained from a single-hidden-layer neural network.*

**Lemma 2.** *The coefficients of the series $P_1$ and $P_2$ can be obtained by the following 1D integrals.*

$$[P_1(d, t)]_{n_1} = \frac{1}{\zeta} \int_{\mathbb{R}} (\lambda_1 - \eta' + \zeta)^d \lambda_1^{n_1 - 1} d\mu_1(\lambda_1)$$

$$[P_2(d, t)]_{n_2} = \frac{1}{\zeta} \int_{\mathbb{R}} (\lambda_2 - \eta + \zeta)^d \lambda_2^{n_2 - 1} d\mu_2(\lambda_2)$$

(S22)

The proof follows by simply plugging-in the fractional binomial expansion inside each of the integrals and verifying that the corresponding two equations for $P_1$ and $P_2$ are indeed equal. The sums over $n$, $n_1$, and $n_2$ are now trivial,

$$2P(t) = P(t; \eta, \zeta) + P(t; , \eta', \zeta) +$$

$$\sum_{d=1}^{\infty} t^{2(d-1)} \int_{\mathbb{R}} \int_{\mathbb{R}} \frac{t^2(2 - t\lambda_1 - t\lambda_2)(\lambda_1 - \eta' + \zeta)^d(\lambda_2 - \eta + \zeta)^d}{\zeta^2(-1 + t\lambda_1)^{d+1}(-1 + t\lambda_2)^{d+1}} d\mu_1(\lambda_1) d\mu_2(\lambda_2)$$

$$= \sum_{d=0}^{\infty} t^{2(d-1)} \int_{\mathbb{R}} \int_{\mathbb{R}} \frac{t^2(2 - t\lambda_1 - t\lambda_2)(\lambda_1 - \eta' + \zeta)^d(\lambda_2 - \eta + \zeta)^d}{\zeta^2(-1 + t\lambda_1)^{d+1}(-1 + t\lambda_2)^{d+1}} d\mu_1(\lambda_1) d\mu_2(\lambda_2)$$

$$= \int_{\mathbb{R}} \int_{\mathbb{R}} \frac{2 - t(\lambda_1 + \lambda_2)}{\zeta^2 \left( \eta t^2(\lambda_1 - \eta' + \zeta) + \eta' t^2(\zeta + \lambda_2) - (1 + \zeta t)(-1 + \zeta t + t(\lambda_1 + \lambda_2)) \right)} d\mu_1(\lambda_1) d\mu_2(\lambda_2)$$

(S23)

Simplifying this expression and utilizing eqn. (S16) yields the expression in eqn. (20).

## 4 Proof outline for general case

### 4.1 General Case

In this supplementary subsection, we remove the assumption that $f$ is required to be linear. In the linear case, in eq. (16), we could directly compute the traces by applying the "mixed-product property" of Kronecker products to the expressions in eq. (15). Moreover, summing the resulting

series to obtain the Stieltjes transform was possible because the individual traces corresponded to Catalan numbers for which a generating function is known. In general, there is no analogous mixed product property to simplify the trace calculations, and we believe that the resulting series does not support a characterization with a single dimensional elliptic integral. With that caveat, we now proceed with the general case.

Our first task is to (asymptotically) evaluate traces of the form:

$$\sum_{\substack{i_1,\ldots,i_{2k}\in\{0,1\}\\i_1+\ldots+i_{2k}=k}} \mathrm{tr}\left[M^{(1)^{i_1}}M^{(2)^{i_2}}\ldots M^{(1)^{i_{2k-1}}}M^{(2)^{i_{2k}}}\right], \tag{S24}$$

where $M^{(1)} = J^{(1)^T}J^{(1)}$ and $M^{(2)} = J^{(2)^T}J^{(2)}$.

Suppose that there are $m$ examples[1], with each example $i$ indexed by $\mu_i$. For a given value of $k$, any trace as in eq. (S24), eventually consists of a sum over a product of per-example Jacobian matrices $J_{\mu_i}^{(1)}$ and $J_{\mu_i}^{(2)}$. Observe that,

$$
\begin{aligned}
\mathrm{tr}\, H^{(0)^k} = \quad &\mathrm{tr}\, \frac{1}{m^k}\sum_{\vec{\mu}}\Bigg[\left([J_{\mu_1}^{(1)}J_{\mu_1}^{(2)}][J_{\mu_1}^{(1)}J_{\mu_1}^{(2)}]^T\right)\\
&\times\left([J_{\mu_2}^{(1)}J_{\mu_2}^{(2)}][J_{\mu_2}^{(1)}J_{\mu_2}^{(2)}]^T\right)\\
&\ldots\left([J_{\mu_k}^{(1)}J_{\mu_k}^{(2)}][J_{\mu_k}^{(1)}J_{\mu_k}^{(2)}]^T\right)\Bigg],
\end{aligned}
\tag{S25}
$$

where $\mu_1 = \mu_k$ (by definition of the trace). For $n, m \to \infty$, where we take the limit over $m$ first and then over $n$,[2] observe that the $\mu_i$s are all pairwise unequal, except that $\mu_1 = \mu_k$ as required. Of course, $n$ and $m$ are equal, by assumption.

Focusing on each term in the previous sum and expanding the "$2 \times 2$" (block) matrices in the $J$s, we get traces over terms of the form:

$$\mathrm{tr}\prod_{i=1}^d J_{\mu_i}^{(b_i)}J^{(b_i)^T}_{\mu_i}, \tag{S26}$$

where $b_i \in \{1,2\}$ and $\mu_i$s are unequal. By the cyclicity of the trace, we can rotate the last Jacobian to the front to re-pair terms and rewrite the above trace as:

$$\mathrm{tr}\prod_{i=1}^d J^{(b_{i+1})^T}_{\mu_{i+1}}J_{\mu_i}^{(b_i)}, \tag{S27}$$

where addition in the $\mu_i$ subscripts is such that $d+1 \mapsto 1$. Finally, expanding the Jacobians into their constituent entries, using equations:

$$J^{(1)}_{ab,i\mu} = W^{(2)}_{ia}f'\Big(\sum_k W^{(1)}_{ak}x_{k\mu}\Big)x_{b\mu} \tag{S28}$$

$$J^{(2)}_{cd,j\nu} = \delta_{cj}f\Big(\sum_l W^{(1)}_{dl}x_{l\nu}\Big). \tag{S29}$$

we can write down each trace as in eq. (S27) as a polynomial in terms of the weights in $W$, the data points in $X$ and $f$. The set of subindices occuring within the polynomial have some cyclic symmetries (follows from the cyclic arrangement of the Jacobians, as above). For example, in the trace calculations below:

$$\mathrm{tr}\left[M^{(1)}M^{(1)}M^{(1)}\right] \;=\; \sum_{abi\mu}\Bigg[W^{(2)}_{i_1a_1}W^{(2)}_{i_2a_1}W^{(2)}_{i_2a_2}W^{(2)}_{i_3a_2}W^{(2)}_{i_3a_3}W^{(2)}_{i_1a_3}$$

$$\times f'\!\left(z_{a_3\mu_3}\right)f'\!\left(z_{a_3\mu_1}\right)f'\!\left(z_{a_1\mu_1}\right)f'\!\left(z_{a_1\mu_2}\right)f'\!\left(z_{a_2\mu_2}\right)f'\!\left(z_{a_2\mu_3}\right)$$

$$\times x_{b_1\mu_1}x_{b_1\mu_2}x_{b_2\mu_2}x_{b_2\mu_3}x_{b_3\mu_3}x_{b_3\mu_1}\Bigg]$$

$$\mathrm{tr}\left[M^{(2)}M^{(2)}M^{(2)}\right] \;=\; \sum_{i\mu d}\Big[\delta_{i_1i_1}f(z_{d_1\mu_1})f(z_{d_1\mu_2})f(z_{d_2\mu_2})f(z_{d_2\mu_3})f(z_{d_3\mu_3})f(z_{d_3\mu_1})\Big]$$

$$\mathrm{tr}\left[M^{(1)}M^{(2)}\right] \;=\; \sum_{abdi\mu}\Big[W^{(2)}_{i_1a_1}W^{(2)}_{i_1a_1}f'(z_{a_1\mu_1})f'(z_{a_1\mu_2})x_{b_1\mu_1}x_{b_1\mu_2}f(z_{d_1\mu_2})f(z_{d_1\mu_1})\Big]$$

$$\mathrm{tr}\left[M^{(1)}M^{(2)}M^{(1)}M^{(2)}\right] \;=\; \sum_{abdi}\Big[W^{(2)}_{i_1a_1}W^{(2)}_{i_2a_1}W^{(2)}_{i_2a_2}W^{(2)}_{i_1a_2}f'(z_{a_1\mu_1})f'(z_{a_1\mu_2})f'(z_{a_2\mu_3})f'(z_{a_2\mu_4})$$

$$\times x_{b_1\mu_1}x_{b_1\mu_2}x_{b_2\mu_3}x_{b_2\mu_4}f(z_{d_1\mu_2})f(z_{d_1\mu_3})f(z_{d_2\mu_4})f(z_{d_2\mu_1})\Big], \quad \text{(S30)}$$

note that the subscript indices in $i$, $a$, $b$ and $\mu$ have some cyclic symmetry.

Note that the polynomial is not necessarily multilinear because there may be identifications between various indices.[3] Similarly, the polynomial is not completely symmetric because of restrictions on the indices induced by matrix multiplication and chain rule for taking derivatives. So, the cyclic symmetries are not completely trivial. Still, the structure induced among the indices is key to evaluating the trace. We map this structure to certain outer-planar graphs (as in [13]) and follow their machinery in evaluating the asymptotic expression for the trace. The latter effectively means that certain analytic details, like computing the saddle point asymptotic approximations can be hidden under the carpet.

Recall that the normalized trace, that we need to evaluate for the moment method, is of the form:

$$\mathbb{E}\,\frac{1}{n_1}\,\mathrm{tr}\,M^k, \qquad\qquad\qquad \text{(S31)}$$

where the matrix $M := f(WX)$, $f$ applied point-wise, and the weights $W$ and input $X$ are Gaussian distributed i.i.d. variables. The crux of the argument that we need from Section 4 of [13], is that the normalized trace, can be written as the integral:

$$\int \Big[f\big(\textstyle\sum_l W_{i_1l}X_{l\mu_1}\big)f\big(\textstyle\sum_l W_{i_2l}X_{l\mu_1}\big)\cdots f\big(\textstyle\sum_l W_{i_kl}X_{l\mu_k}\big)$$

$$\times f\big(\textstyle\sum_l W_{i_1l}X_{l\mu_k}\big)\Big]\,\mathcal{D}W\mathcal{D}X. \qquad\qquad \text{(S32)}$$

After introducing auxiliary matrix valued variables $Z$ and $\Lambda$, evaluating the $X$ and $W$ integrals, they are able to simplify the last integral to:

$$\int \Big[\exp\big[-\tfrac{n}{2}\log\det|1+\tfrac{1}{n}\Lambda\Lambda^T|-i\,\mathrm{tr}\,\Lambda Z\big]$$

$$\times f(Z_{i_1\mu_1})...f(Z_{i_1\mu_k})\Big]\mathcal{D}\lambda\mathcal{D}z\,, \qquad\qquad \text{(S33)}$$

where $\mathcal{D}\lambda = \prod_{\lambda_{\alpha\beta}\in\Lambda}\frac{d\lambda_{\alpha\beta}}{2\pi}$ and $\mathcal{D}z = \prod_{z_{\alpha\beta}\in Z}dz_{\alpha\beta}$. Finally, using saddle point approximations near the origin (the Gaussians are all mean zero), allows them to evaluate the last integral, and therefore the normalized trace, asymptotically as a polynomial in terms of $\eta$, $\zeta$ (the same $\eta$ and $\zeta$ in our main result). And, they also compute the Stieltjes transform of $M$.

However, unlike [13], we have two matrices $M_1$ and $M_2$ and in order to evaluate traces of the form:

$$\sum_{\substack{i_1,\ldots,i_{2k}\in\{0,1\}\\ i_1+\ldots+i_{2k}=k}} \mathrm{tr}\left[M^{(1)^{i_1}}M^{(2)^{i_2}}...M^{(1)^{i_{2k-1}}}M^{(2)^{i_{2k}}}\right], \qquad \text{(S34)}$$

Figure S1: The configuration of $a$ and $d$ indices for $\mathrm{tr}(M_1 M_2 M_1 M_2)$ is shown. In general, each green and blue arc corresponds to a admissible graph [13], connected by a single edge, corresponding to the indices of the common variable.

the resulting integrals as in eq. S32 are replaced by those of the form (cf. eq. S30):

$$\sum_{abdi} \int W^{(2)}_{i_1 a_1} W^{(2)}_{i_2 a_1} W^{(2)}_{i_2 a_2} W^{(2)}_{i_1 a_2} f'(Z_{a_1 \mu_1}) f'(Z_{a_1 \mu_2}) f'(Z_{a_2 \mu_3}) f'(Z_{a_2 \mu_4})$$
$$\times \quad X_{b_1 \mu_1} X_{b_1 \mu_2} X_{b_2 \mu_3} X_{b_2 \mu_4} f(Z_{d_1 \mu_2}) f(Z_{d_1 \mu_3}) f(z_{d_2 \mu_4}) f(Z_{d_2 \mu_1}) \mathcal{D} W \mathcal{D} X \,, \quad \text{(S35)}$$

where the matrix $Z = WX$. A slightly more complicated scenario. On the other hand, we assume that all our matrices, weights as well as inputs are square i.e., dimension $n \times n$, and we assume unit variance throughout, which helps simplify the situation a little.

The crux of the question, when trying to evaluate an integral of the form S35 is what is the relative contribution of terms where certain sets of subscript indices, and therefore coefficients, are identified? What kind of terms dominate the expression when calculating the asymptotic value of the trace? What kind of identifications lead to sub-leading terms?

For example, the following lemma shows that when evaluating $\mathrm{tr}\, M_1^k$ for $k \geq 3$, all $b$ indices must be equal or else the trace is asymptotically zero. The reason being that the underlying covariances of $X_{b\mu}$ and $X_{b'\mu'}$ are zero when $b \neq b'$.

**Lemma 3.** *Given an expansion of $\mathrm{tr}\, M_1^k$ in terms of the entries of $W$ and $X$, the left indices of the $X$ terms i.e., the $b_i s$ in $\prod_i X_{b_i \mu_i}$, are either all equal, or the contribution to the trace is zero. Furthermore, the statement also holds for each run of $M_1 s$ in eq. S34.*

The above is a structural result for the $b$ indices. Similarly, consider the $a$ and the $d$ indices in eq. S30. We can arrange them as vertices of a cyclic graph to obtain a two-colored cycle corresponding to the $a$ and $d$ indices in $\mathrm{tr}(M_1 M_2 M_1 M_2)$.

The green arcs correspond to the $a$ indices (coming from the $W$s in $M_1$) and the blue arcs correspond to the $d$ indices (coming from the $Z$ terms in $M_2$s). Note that for a given term, some of the $a$ indices may be equal, in which case those vertices within the green arcs would be identified, and similarly for the vertices / indices in the blue arcs. This identification of vertices results in a complicated graph structure for the blue and green graphs, as opposed to a simple path structure.[4]

The next question can now be framed as follows: Every term arising from the trace corresponds to a graph, so which type of graphs lead to dominant terms i.e., terms that are asymptotically significant?

In [13], it was shown that only terms corresponding to the "admissible graphs", which are graphs consisting of edge disjoint cyclic blocks such that their planar dual forms a tree, contribute asymptotically to the trace in eq. S31.

The asymptotically dominant terms in the trace, corresponding to blue or green graphs still correspond to the *admissible graphs* defined in [13] i.e., the dominant terms will correspond to graphs that can be partitioned into edge-disjoint cyclic blocks, whose planar duals are trees. However, in our case the planar duals of the blue and green graphs, taken separately, may be disconnected (if there are no vertex identifications across arcs), and may therefore form forests. Despite this, the techniques

of [13] are all still applicable and such "admissible graphs" form the leading terms of the trace in eq. S24. We skip the lengthy proof since the idea is the same as that in [13].

Now, keeping the above in mind, consider the evaluation of traces of the form $\text{tr}\left[M^{(1)^{i_1}}M^{(2)^{i_2}}...M^{(1)^{i_{2k-1}}}M^{(2)^{i_{2k}}}\right]$. In particular, consider the trace in eq. S30 and the corresponding integral in eq. S35 as a concrete example. Suppose that there are $2d$ alternations between $M_1$ and $M_2$ runs[5], that the total degree of $M_1$ is $n_1$, and that of $M_2$ is $n_2$. Therefore, $d = 2$, $n_1 = 2$ and $n_2 = 2$ in our concrete example. We then define the following quantities:

- Let $[H(d, \lambda_1, \lambda_2)]_{n_1,n_2}$ denote the number of such monomials terms i.e., those having $d$ alternations, and total degrees $n_1$ and $n_2$ in $M_1$ and $M_2$, respectively.
- Let $P_1(d, t)$ denote the contribution of the $M_1$ terms to the trace. Equivalently, the expected value of the integral corresponding to the trace when any variables with "d" indices i.e., those that belong to the "blue" graph, are dropped.
- Let $P_2(d, t)$ denote the contribution of the $M_2$ terms to the trace. Equivalently, the expected value of the integral corresponding to the trace when any variables with "a" indices, those that belong to the "green" graph, are dropped.

The following three lemmas can then be shown using only elementary methods. The proof idea is similar to that of the proofs in [13].

Proof sketch for the first two lemmas: one assumes that the sequence of blue and green arcs (admissible graphs) on the "circle", comprising of total $n_1$ and $n_2$ vertices is fixed. The proof follows by separating out $i$ (say) out of $d$ "arcs" and holding them to be disconnected i.e., no vertex identifications in-between those arcs. The remaining arcs are assumed to be connected via non-crossing vertex identifications between those arcs. Recall that each arc corresponds to an admissible graph. This eventually leads to the $\frac{d-i-1}{d+i-1}\binom{d+i-1}{i}$ factor in Lemma 4.

So for the proof one needs to count the contribution for each such configuration i.e., $i$ disconnected and $d - i$ connected, of admissible graphs. For each fixed $i$, one effectively has $i + 1$ admissible graphs, and "bubbles" corresponding to contributions of $\zeta$ terms at their boundaries. This corresponds to the $\frac{1}{(1-P(t;\eta',\zeta))^{d-i-1}}$ term in Lemma 4.

Essentially, the proof of Lemma 4 is just a direct extension of the proof in [13], where the calculation is for a configuration consisting of one admissible graph, as opposed to $d$ disjoint admissible graphs.

Note that counting the number of ways of selecting the $i$ connected graphs is similar to (but not the same as) counting the number of non-crossing dissections of a $(i + 2)$-gon. The latter, however, has a bijection to a standard Young's tableaux (cf. [17]); while in our case, vertex identifications do not formally lead to lines but to "cyclic blocks" (cf. [13]), and that leads to a subtle, but asymptotically significant, difference in the final calculated value. This comprises the essential outline of the proofs of the two lemmas below.

**Lemma 4.**

$$P_1(d, t) = \zeta^{d-1} \sum_{i=0}^{d-1} \frac{\left(\frac{d-i-1}{d+i-1}\binom{d+i-1}{i}\right)}{(1 - P(t; \eta', \zeta))^{d-i-1}} \tag{S36}$$

Similarly, we have the lemma below.

**Lemma 5.**

$$P_2(d, t) = \zeta^{d-1} \sum_{i=0}^{d-1} \frac{\left(\frac{d-i-1}{d+i-1}\binom{d+i-1}{i}\right)}{(1 - P(t; \eta, \zeta))^{d-i-1}} \tag{S37}$$

Finally, in the third lemma, one counts the number of configurations of blue and green arcs, given the total degree $n$. The proof of which is elementary combinatorics. It consists of simply counting the number of ways of interlacing $d$ blue and $d$ green "arcs", with blue arcs covering $n_1$ points and green arcs covering $n_2$ points on a circle with $n$ points. Therefore, the corresponding generating function $H$ in $[H(d, \lambda_1, \lambda_2)]_{n_1,n_2}$ is given by the following lemma.

**Lemma 6.**

$$H(d, \lambda_1, \lambda_2) = \frac{2\lambda_1\lambda_2 - \lambda_1^2\lambda_2 - \lambda_1\lambda_2^2}{(-1+\lambda_1)^{d+1}(-1+\lambda_2)^{d+1}}. \tag{S38}$$

Recall that, one only needs to ensure that each color (blue and green) graph is a disjoint union of admissible graphs. Also recall that, evaluating the expectation integral over an admissible graph, can be expressed only in terms of the number of cyclic blocks in the block structure of the admissible graph. Therefore, given the number of alternations $d$, and the number of vertices in the blue and green graphs, the generating function $P(t)$ can be written as a sum over the multiplication of the three generating functions as in Eqn. S17! The above sketches the proof of Lemma 1.

## Footnotes

[1] In fact, we will assume $n = m$ i.e., we assume the width of the network and the number of examples both go to infinity at exactly the same rate. This way all our matrices are square, and our calculations are simplified.

[2] This is indeed the case when computing the limiting spectrum for the Fisher.

[3]Except in the case of $\mu$s as mentioned above.

[4] So the figure of a circle with arcs is deceptively simple!

[5] Assume that the terms $M^{(1)^{i_1}}M^{(2)^{i_2}}...M^{(1)^{i_{2k-1}}}M^{(2)^{i_{2k}}}$ are laid out in a circle.