[Reviews · NeurIPS 2018]

Reviewer 1



1) Summary This paper computes the spectrum of the Fisher information matrix of a single hidden-layer neural network with squared loss, under two main assumptions: i) the number of hidden units is large and ii) the weights are Gaussian. Some further work is dedicated to a conditioning measure of this matrix. It is argued that neural networks with linear activation function suffer worse conditioning than those with non-linearities. 2) Major comments I am overall very impressed by this article. As the Pennington and Worah NIPS 2017 paper cited in the article, it successfully uses random matrix theory to analyze the behavior of neural networks, but definitely going one step further. Even though the model is simple, the results are rigorous and quantitative. My main concerns are about the assumptions made in the main result. Detailed comments follow. - While I find the argument ``consider the weights as random'' (ll. 52--58) very convincing, I think that the Gaussian assumption is quite restrictive. Surely, it is valid during the first step of the training (especially when the weights are initialized at random as Gaussian random variables). But my feeling is that after a very short time training the network, the weights are no longer Gaussian. More precisely, while the `randomness' could very well be Gaussian, some structure should emerge and the zero-mean assumption should fail. Do the authors have any comments / experiments regarding the evolution of the distribution of the weights during the training? - A very related question: keeping the Gaussian distribution as a first step, I am curious as to how robust is the proof mechanism to introducing some mean / variance structures in the distribution of the weights? - Th. 1 is very hard to read at first sight and Section 2.5 is very welcome. Is it possible to prove another intermediary result / corollary for a non-linear activation function that is also explicit? - Reading the statement of Th. 1, it seems that f needs to be differentiable. In particular, ReLU-like activation functions would be excluded from the present analysis. However, srelu_0 is investigated in Table 1. Can you clarify (even briefly) what are the regularity assumption one needs on f, if any? - Reading the proof of the main result, I had a lot of trouble figuring the graph structure associated with the indices in the trace. As the article puts it, ``the figure of a circle with arcs is deceptively simple.'' I suggest to provide another example for a slightly more involved example than \trace{M_1M_2M_1M_2} in order to help the interested reader. - A lot of details from the proof of the main result are missing, often pointing to the Pennington and Worah NIPS 2017 paper (in particular lemmas 4, 5, and 6). While I trust that the maths are correct, I think that the article should be selfsufficient, especially since there is no limit on the size of the supplementary material. 3) Minor comments and typos - l. 81: Eq. (3): I think there is a missing Y in the law under which the expectation is taken - l. 83: J^\top J should read \expec{J^\top J} - l. 103: Eq. (5) vs Eq. (8): please be consistent with your convention for the integral - l. 111: Eq. (8): the far right-hand side can be confusing. I would suggest adding a pair of parentheses. - l. 160: ``it's'' -> it is - l. 189: Eq. (27): missing punctuation - l. 335: the definition of \mu_1 and \mu_2 is a bit far at this point. - l. 388: Figure S1: missing index, M should read M_2 on the right side of the figure.

Reviewer 2



This submission is a theoretical contribution on the spectrum of Fisher information matrix of deep neural networks. Based on random matrix theory, the authors studied such a spectrum in a very simplified setting: a one-hidden layer feed-forward network, where both the inputs and all neuron network weights are i.i.d. Gaussian distributed, all layers have the same width, the activation function has zero Gaussian mean and finite Gaussian moments, and the cost function is the squared loss. The main results are summarized in theorem 1: in the limit case (very wide layer), the Stieltjes transform of the spectral density is in closed form. The author further showed consistency of the theoretic prediction of the spectrum density with numerical simulations using several activation functions (including the linear case), which shows perfect agreement. The theoretical results present a solid advancement on the thread of applying random matrix theory to deep learning. It is meaningful in understanding the Fisher information matrix of neural networks. While I am convinced that this paper should be accepted by NIPS as a nice theoretical contribution, I am not familiar with random matrix theory as indicated by the confidence score. The main comments I have is regarding the very constrained assumptions and the writing. It is true that it is difficult to get elegant results as theorem 1 for deep learning without making simplified assumptions. However, in the end, the authors did not make any attempt on generalizing the results to more general settings, or providing relevant discussions, or making numerical simulations to show the (in)consistency of the spectrum when those assumptions fail. Therefore the results have limited values and may not have a large impact. As a relevant question, is it possible to extend the results to the case, where dim(X) is fixed while the size of the hidden layer goes to infinity? If this case can be provided in section 3, the contribution will be much more systematical. Did you make any experimental study on the case, where the constraints are slightly violated, while the prediction still roughly holds? If such results can be presented as figures in section 3/4, it will make the contribution more stereoscopic. Regarding the writing, the overall technical and English quality is good. The introduction is a bit too long and deviating from the main contributions. The authors raised some big questions, such as "under which conditions first-order methods will work well". These questions are not really answered at the end. The authors are therefore suggested to streamline section 1 and "cut to" the topic. It needs less words to motivate that the spectrum of the Fisher/Hessian is important. Instead, use the saved space to provide more discussion in the end on generalizing the results. Please see the following minor comments for more details. Minor comments: Through out the paper there are some extra commas just before the equations. For example, L80 "can be written as , H=H^0+H^1". Please use parentheses or brackets to denote the expectations as E_X(...). The current notation looks a bit weird. In equation (1) there is an expectation with respect to the random variable Y. In the following equations it is gone. Please check carefully and be consistent. Use a different symbol such as ":=" for definitions. L101 limiting spectral density can be accompanied with an equation. Please try to keep the paper self-contained and refer a bit less to the SM. equation (29) can be moved to an earlier place to introduce f_opt Figure(2,b): please make clear which curves are predictions and which curves are simulations A relevant (unpublished) work is "Universal Statistics of Fisher Information in Deep Neural Networks: Mean Field Approach. Ryo Karakida, Shotaro Akaho, Shun-ichi Amari. 2018.". As this work is unpublished, it is only for the authors' interest to have a look. -After rebuttal- Thank you for your rebuttal. Please try include related discussions on generalizations (e.g. in the last section).

Reviewer 3



The authors investigate the impact of local curvature of the loss surface to model optimization. They do this by computing the eigenvalues of the Fisher information matrix in the context of a neural network with a single hidden layer with Gaussian data and weights. The authors find that nonlinear networks are better conditioned than linear networks and it is possible to tune the non-linearity in such a way to speed up first-order optimization. The paper is well written with a lot of attention to detail. The derivation is well explained and the figures are useful. The empirical evaluation is interesting and novel. In general I think the paper is a useful extension of Pennington/Worah (2017) and a contribution to the conference. My main concern is that the empirical evaluation is limited. The analysis of the nonlinearity is interesting, but limited in scope. I would be curious if the method can also give insight into optimizers (e.g. momentum, learning rate, or fancier such as adam). Nit: Line 93: spurious "a"